# Atomically unveiling the structure-activity relationship of biomacromolecule-metal-organic frameworks symbiotic crystal

Linjing Tong[1], Siming Huang [2], Yujian Shen[1], Suya Liu[3], Xiaomin Ma[4], Fang Zhu[1], Guosheng Chen [1✉] & Gangfeng Ouyang [1]

Crystallization of biomacromolecules-metal-organic frameworks (BMOFs) allows for orderly assemble of symbiotic hybrids with desirable biological and chemical functions in one voxel. The structure-activity relationship of this symbiotic crystal, however, is still blurred. Here, we directly identify the atomic-level structure of BMOFs, using the integrated differential phase contrast-scanning transmission electron microscopy, cryo-electron microscopy and x-ray absorption fine structure techniques. We discover an obvious difference in the nanoarchitecture of BMOFs under different crystallization pathways that was previously not seen. In addition, we find the nanoarchitecture significantly affects the bioactivity of the BMOFs. This work gives an important insight into the structure-activity relationship of BMOFs synthesized in different scenarios, and may act as a guide to engineer next-generation materials with excellent biological and chemical functions.

[1] MOE Key Laboratory of Bioinorganic and Synthetic Chemistry, School of Chemistry, Sun Yat-sen University, Guangzhou 510275, China. [2] School of Pharmaceutical Sciences, Guangzhou Medical University, Guangzhou 511436, China. [3] Shanghai Nanoport, Thermo Fisher Scientific, Jinke Road. Pudong District, Shanghai 200120, China. [4] Cryo-EM Center, Southern University of Science and Technology, Shenzhen 518055, China.
✉email: chengsh39@mail.sysu.edu.cn

Crystallization of biomacromolecules-metal-organic frameworks (BMOFs) composites is an advanced biotechnology that allows to orderly assemble symbiotic hybrid with desirable biological and chemical functions[1–5]. The formed symbiotic crystal well inherits the MOFs crystallographic structures and possesses structurally matched 3D microenvironment for biomacromolecules confinement[6–9]. The internalized biomacromolecules in a BMOFs crystal have shown significantly enhancement on stability and reusability, and the pore size-tailorable BMOFs carriers provide a gatekeeper-like effect for guest sieving[10–13]. This symbiotic crystallization technique brings an insight into the development of next-generation materials in drug delivery[14,15], chemical and biological sensing[16], and biomimetic catalysis[2].

The large-scale applications of this biocrystal in the aforementioned fields, however, remain substantial challenges. One of the tough problems, that is an urgent need to dissect, is the structure-activity relationship[4,16–19]. To our best knowledge, zeolitic imidazolate frameworks-8 (ZIF-8), featured with the biocompatible assembly conditions (room temperature and aqueous phase etc.) that shield biomacromolecules against denaturation during the symbiotic crystallization, is the favored MOFs matrix[9–12,14–19]. Unfortunately, the activities of biomacromolecules-ZIF-8 (BZIF-8) crystals are highly controversial. In theory, the narrow aperture of ZIF-8 (ca. 3.4 Å) decides that very limited guests are accessible to the internalized biomacromolecules, and the activity of biomacromolecules after crystallization was observed to be partially or even completely restrained[16–19]. Strangely, in some cases, the internalized biomacromolecules trended to maintain comparable bioactivity as the free counterpart, and even unexpectedly carried out the biocatalysis process where the enzymatic substrate has a large size than the crystallographic pore structure of ZIF-8[2,4,20,21]. To resolve these controversial views, in depth unveiling the microstructure of BZIF-8 crystal in different assembly scenarios is the essential task, yet, still remains unknown.

Here, we pioneered to use the integrated differential phase contrast-scanning transmission electron microscope (iDPC-STEM), cryo-electron microscopy (cryo-EM) and X-ray absorption fine structure (XAFS) techniques to investigate the microstructure of BZIF-8 composites crystallized in different pathways. The atomic structure of BZIF-8 composites are directly identified. These atomic-level information provide meaningful insights into the significant activity difference of ZIF-8 biohybrid synthesized in different scenarios, and may guide to engineer next-generation materials with excellent biological and chemical functions.

## Results

**Biomacromolecules-ZIF-8 synthesis and bioactivity characterization.** A recent work by Patterson and co-workers utilized lattice-resolution cryo-EM to study the nucleation process of proteins-ZIF-8[22], and the results found that the proteins-ZIF-8 mainly crystallized through two different pathways, which was depended on the 2- Methylimidazole (HmIM): $Zn^{2+}$ precursor ratios: (1) solid-state transformation. In this pathway, the biomacromolecules amorphous phase undergo a solid-state transformation at the spontaneously growing ZIF-8 crystal surface through heterogeneous crystallization (the composites denoted as BZIF-8-S herein); (2) biomacromolecules-induced crystallization. In this pathway, the biomacromolecules triggered the ZIF-8 crystallization around their surfaces directly via electrostatic interaction (the composites denoted as BZIF-8-B herein). However, to our best knowledge, the high-resolution structure of BZIF-8 still remains unknown, which significantly limits the understanding of the structure-activity relationship of this symbiotic crystal.

We herein attempt to explore this structure-activity relationship via imaging the atomic-level structure of BZIF-8 crystallized in different pathways. We synthesized BZIF-8-S and BZIF-8-B using glucose oxidase (GOx) as biomacromolecules (Fig. 1a). In the high precursor ratio (1.2 M HmIM and 0.1 M $Zn^{2+}$), the nucleation of ZIF-8 crystals was slow and spontaneous, and BZIF-8 was formed by solid-state transformation (Supplementary Fig. 1). While in the low precursor ratio (160 mM HmIM and 40 mM $Zn^{2+}$), the HmIM/Zn precursor was unable to crystallize, and the proteins triggered the ZIF-8 nucleation directly (Supplementary Fig. 2). The PXRD showed that both of the BZIF-8-S and BZIF-8-B inherited the *SOD*-type crystallographic structure (Fig. 1b), and the typical amide I band that assigned to the polypeptide skeleton of proteins, was recorded in the Fourier transform-infrared spectrum (FTIR), confirmed the internalization of GOx (Supplementary Fig. 3). In addition, the thermogravimetric analysis (Supplementary Fig. 4) and $N_2$ adsorption isothermal curve (Supplementary Fig. 5) also supported the GOx incorporation. We evaluated the GOx loading through standard Bradford assay (Supplementary Table 1) and examined the bioactivity of BZIF-8 in terms of identical GOx dosage (Fig. 1c). An interesting phenomenon was that the bioactivity between BZIF-8-B and BZIF-8-S was quite different. The bioactivity of BZIF-8-B was comparable to the free counterpart, whereas the activity of BZIF-8-S was significantly inhibited (Fig. 1c). Such a difference on bioactivity between BZIF-8-B and BZIF-8-S is

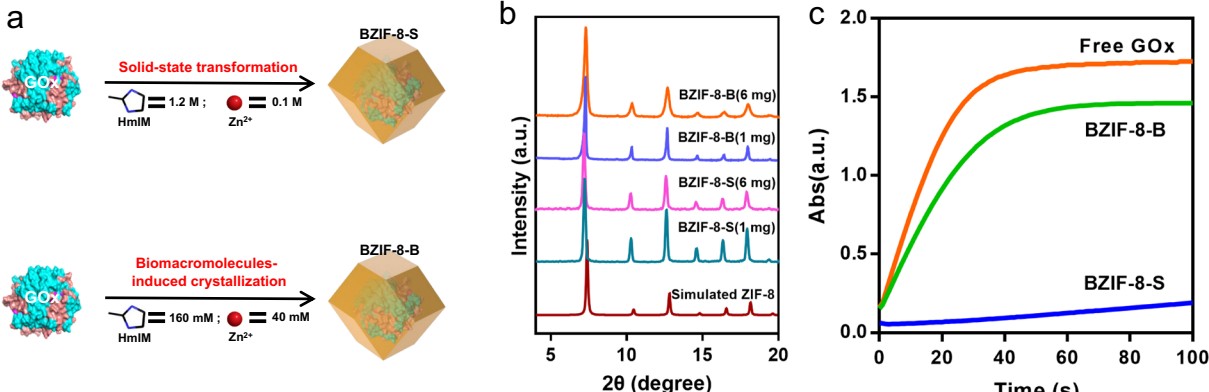

**Fig. 1 BZIF-8 synthesis and characterization. a** Schematic illustration of the Biomacromolecules-ZIF-8 composites crystallized through solid-state transformation and biomacromolecules-induced crystallization. HmIM: 2-Methylimidazole. **b** The PXRD the typical BZIF-8 crystals. The numbers in brackets represent the amount of biomacromolecule used in the crystallization. **c** The bioactivity of BZIF-8 crystals when using 8 mg GOx dosage in the crystallization.

ubiquitous, regardless of the amount of enzyme used in the synthesis (Supplementary Fig. 6).

**Atomic-level structure imaging**. We affirmed that the HmIM and $Zn^{2+}$ used in our scenarios could not disturb the bioactivity of GOx (Supplementary Fig. 7), and also verified the folded structures of GOx were maintained after the crystallization by fluorescence profile (Supplementary Fig. 8). In addition, we designed another experiment to confirm that the particle size has limited effect on the activity of BZIF-8-S (Supplementary Fig. 9). We next attempted to profile the high-resolution structure of BZIF-8, and aimed to understand the structure-activity relationship. All of the BZIF-8 samples used for the subsequent characterization studies are crystallized using 8 mg enzymes, unless otherwise statement. The PXRD result reflects the macroscopic phase of the crystallization system, and is difficult to clearly identify the microstructures such as detective and amorphous phases (Fig. 1b). Seeing is believing. The recently developed Low-electron-dose TEM technique allows to profile the microstructure of the electron beam-sensitive materials[23,24], and such microstructure is difficult to be obtained by PXRD. ZIF-8 could well maintain its intact crystallographic structure after exposure to 50 $e^-/Å^2$ at low temperature[25]. We optimized the cumulative electron dose from 7 to 60 $e^-/Å^2$, and found that the high imaging quality was acquired when using a cumulative electron dose of 30 $e^-/Å^2$ (Supplementary Fig. 10). Using cryo-EM, abundant structural information of biomacromolecules-ZIF-8 composites that previously not seen at room temperature were observed (Fig. 2a). The sequential hexagonal honeycomb lattice along the ⟨1 1 1⟩ direction was profiled in all of BZIF-8-S composites, indicating the intact crystalline structure (Figs. 2a, b). For further insight into the structure of BZIF-8-S, the emergent integrated differential phase contrast-scanning transmission electron microscope (iDPC-STEM) technique[26–28], which enables to image the beam-sensitive materials with high resolution and signal-to-noise ratio, was applied. We randomly imaged six BZIF-8-S particle using iDPC-STEM with lower than 1 pA electron-beam current. In the overfocal imaging mode, the atomic structure of BZIF-8-S was clearly unraveled from the [111] (Fig. 2c and Supplementary Fig. 11) and [100] (Fig. 2d and Supplementary Fig. 12) projection, wherein the bright spots in the lattice correspond to empty pore space. Specially, the periodic cavity with ca. 12 Å formed by Zn clusters and HmIM can be directly identified throughout the framework, which is well-matched with the theoretical structure (Fig. 2c, d and Supplementary Figs. 11 and 12). This cavity has ca. 3.4 Å 6-ring window (along the [111] projection) and 2.8 Å 4-ring window (along the [100] direction), respectively. Such an intact crystallographic structure with narrow pore windows would limit the diffusion of catalytic substrate that decreasing the bioactivity (Fig. 1c and Supplementary Fig. 6).

To lift the veil of the high bioactivity of BZIF-8-B, the high-resolution structure of BZIF-8-B was also carefully profiled based on cryo-EM and iDPC-STEM. The lattice-resolution images were also obtained under the same cumulative electron dose using cryo-EM (Fig. 3a). Different from BZIF-8-S, unsharp lattice and weak diffracted intensity in corresponding fast Fourier transform (FFT) were seen in the BZIF-8-B crystal (Fig. 3b). In addition, we observed that the lattice was inconsecutive (Figs. 3b, c), indicating the amorphous phase was formed in the crystallization process of BZIF-8-B. For insight into the atomic structure of BZIF-8-B, iDPC-STEM was also carried out. In the underfocal imaging mode, we randomly imaged five BZIF-8-B particle with atomic resolution, wherein the dark spots in the lattice correspond to empty pore space (Fig. 3d and Supplementary Figs. 13–16). Viewing from the [111] projection, the interrupted hexagonal

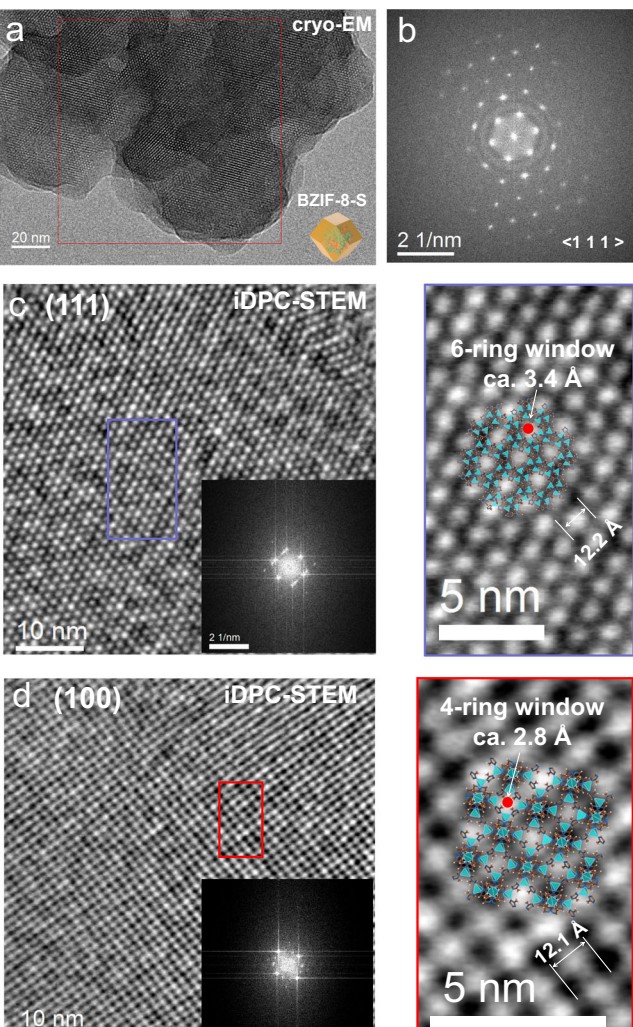

**Fig. 2 High-resolution structure imaging of BZIF-8-S. a** The cryo-EM structure of BZIF-8-S. **b** The fast Fourier transform pattern of the selected area in (**a**). The iDPC-STEM image of BZIF-8-S from the [111] projection (**c**) and the [100] projection **d** in the overfocal imaging mode, and the structural analysis of the selected area. The insert was the FFT pattern from the corresponding projections.

honeycomb lattice was captured using iDPC-STEM (Fig. 3d). In addition, the intensity in FFT (Fig. 3e) was much higher than that using cryo-EM. It allowed to directly observe the ca. 12 Å hexagonal honeycomb cavities, connected by Zn clusters and HmIM (Fig. 3f). However, such an ordered connection was incontinuous (Fig. 3g, h), in stark contrast to the intact crystalline structure observed in BZIF-8-S. The enlarged region A in Fig. 3d showed the coexistence of crystalline phase and amorphous phase, and the amorphous assembly resulted in the inhomogeneous cavity with large size (Fig. 3g). From the view of the enlarged region B in Fig. 3d, we also imaged abundant coordination defects in some hexagonal honeycomb lattice, with the apparent features of a missing linker or clusters (Fig. 3h). These defects also broadened the pore window that accelerating the entrance of catalytic substrate. It was worth to mention that such abundant amorphous phase and defective structure were widespread in BZIF-8-B (Supplementary Figs. 13–16). The different crystallinity of BZIF-8-B and BZIF-8-S may result from the different crystallization rate. For BZIF-8-B, the biomacromolecules trigger the ZIF-8 crystallization around their surfaces[22],

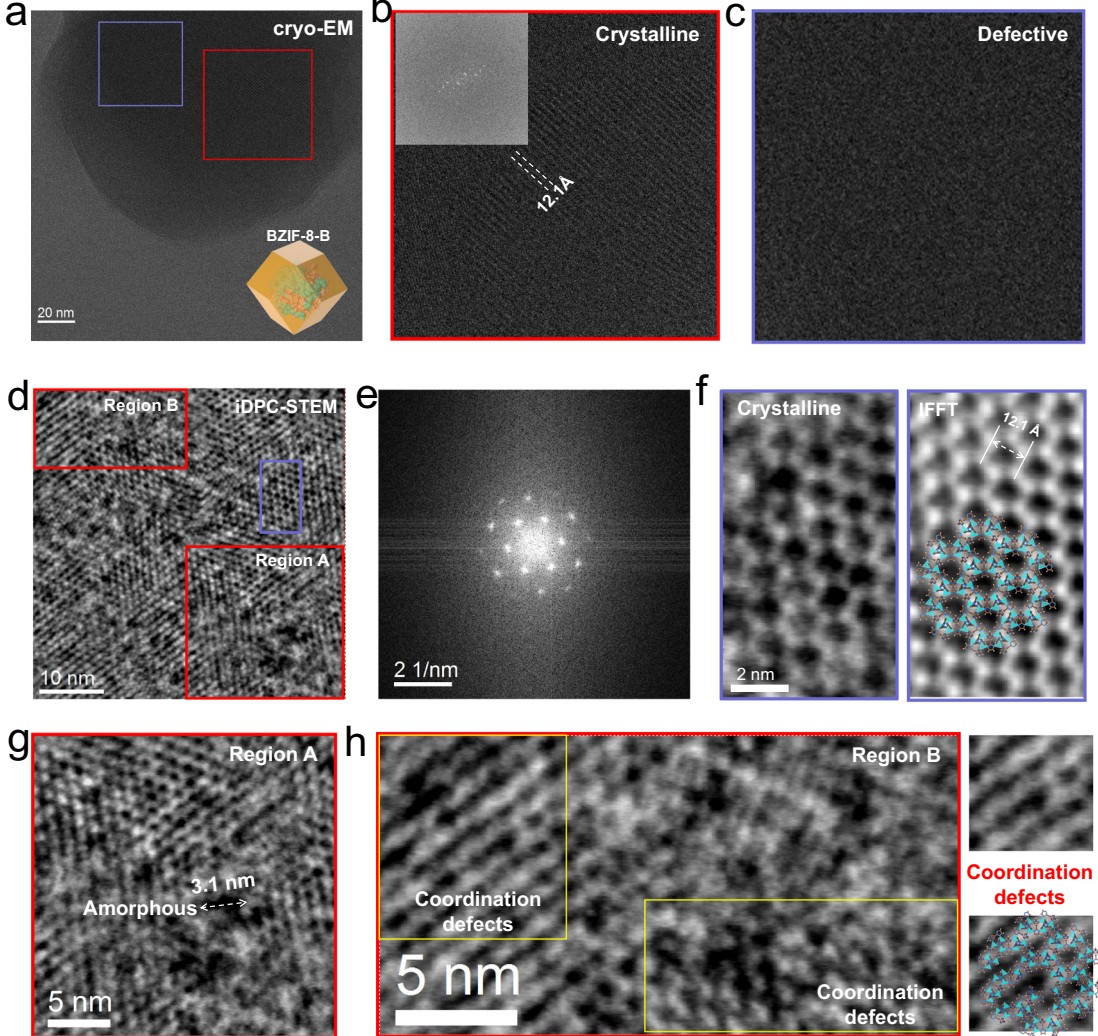

**Fig. 3 High-resolution structure imaging of BZIF-8-B.** The cryo-EM structure (**a**) of BZIF-8-B, and the amplified images of the selected red (**b**) and pink (**c**) area in (**a**). The insert in (**b**) was the FFT pattern. The iDPC-STEM image (**d**) and the corresponding FFT pattern (**e**) of BZIF-8-B from the [111] projection. (**f**) The amplified iDPC-STEM image of the selected pink area (left), and the corresponding inverse FFT (IFFT) pattern (right). It showed the atomic-resolution hexagonal honeycomb cavities. The amplified iDPC-STEM image of the selected red area: Region A (**g**) and Region B (**h**). The detailed analysis presented the amorphous phase and coordination defects.

and the crystallization rate is fast (Supplementary Fig. 1) that results in the poor crystallinity. However, for BZIF-8-S, the biomacromolecules amorphous phase undergoes a solid-state transformation at the spontaneously growing ZIF-8 crystal surface through heterogeneous crystallization[22]. Such crystallization rate is relatively slow, and hence favors for the highly crystalline species growth (Supplementary Fig. 2).

Based on the atomic structure profiles, it is understandable why the BZIF-8-B is highly bioactive (Fig. 1c), because several reports by Ge and co-workers have confirmed the important role of defects in solid supports for retaining high enzyme activity[4,16]. Importantly, we also synthesized these two species of BZIF-8 using other proteins including alcohol dehydrogenase (ADH) and catalase (CAT), and all of the bioactivity assays supported that BZIF-8-B specie possessed the much higher catalytic ability (Supplementary Figs. 17 and 18).

**XAFS spectra**. To give a close examination on the coordination structures, the BZIF-8 were characterized with X-ray absorption fine structure (XAFS), showed in Supplementary Fig. 19. The

peaks at 1.9 Å in the fitted R-space data (Fig. 4a), assigned to the coordination of Zn and N atoms[4], were identified in both of BZIF-8-S and BZIF-8-B. It is suggested that the Zn-N coordination also mainly presented in the biocomposites under these two crystallization pathways. Based on the fitted R-space data, we found that the apparent local coordination number of the Zn atoms in BZIF-8-S is 4.061 (Fig. 4b), which was well in agreement with the theoretical coordination number in *SOD*-type ZIF-8 (the coordination number of the Zn atoms is 4.0). The data indicated the intact crystallographic structure of BZIF-8-S, which was also supported by the high-resolution imaging (Fig. 2 and Supplementary Figs. 11 and 12). As a comparison, the calculated coordination number of the Zn atoms in BZIF-8-B turned out to be 4.272 (Fig. 4b). Such an enhancement on the coordination number of Zn atoms was clearly presented in the wavelet transform spectra from the K-space data (Figs. 4c, d). In the wavelet transform spectra, the intensity maximum in R and K space reflects the coordination number of the Zn atoms, and the contour plot of BZIF-8-B displayed higher intensity maximum in R and K space compared with BZIF-8-S. The higher apparent coordination number of Zn atoms indicated that the loss of Zn

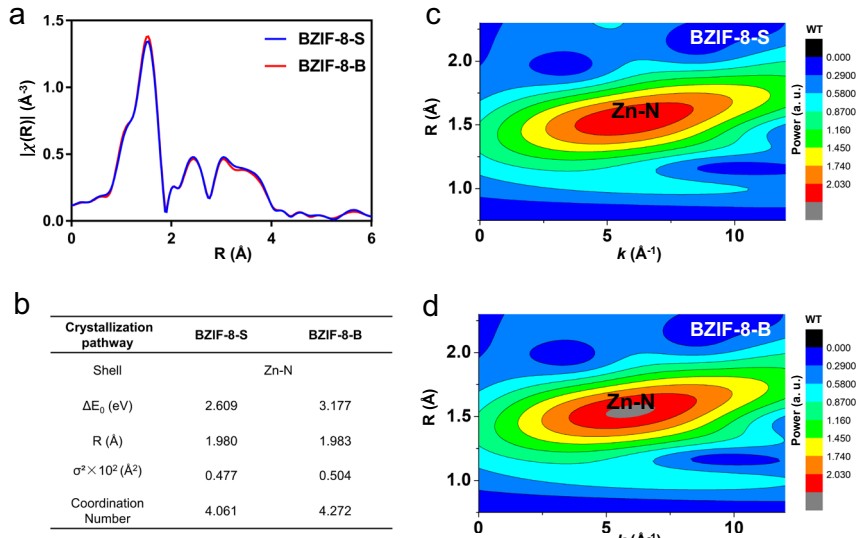

**Fig. 4 Coordination structure profiles. a** The Fourier-transformed (FT)–EXAFS spectra of BZIF-8-B and BZIF-8-S. **b** The data obtained from fitting XAFS spectra to pure ZIF-8 crystal $f_{eff}$ path. $\Delta E_0$: the inner potential correction; R: bond distance; $\sigma^2$: mean square racial displacement, Debye-Waller factor. The wavelet transform spectra of BZIF-8-S (**c**) and BZIF-8-B (**d**) from the K-space data. The intensity maximum in R and K space reflected the coordination number of the Zn atoms. WT means wavelet transform.

atoms, and also suggested the defective structure in BZIF-8-B by biomacromolecules-induced crystallization. This structural defect explored by XAFS was well in agreement with the microstructure profiled by iDPC-STEM (Fig. 3 and Supplementary Figs. 13–16). It is observed that the 6-ring and 4-ring opening windows of ZIF-8-S are only 3.4 Å and 2.8 Å (Fig. 2 and Supplementary Fig. 11 and 12), respectively, which will limit the diffusion of catalytic substrates (the substrate of GOx, that is glucose, is ca. 4.2 × 5.4 Å). The fact that glucose still can activate the GOx, encapsulated within the highly crystalline ZIF-8-S, is attributed to the rotation of the imidazolate rings of ZIF-8 that allow molecules of ca. 5.0 Å to access the pores[29]. Yet, this ZIF-8-S-confined biocatalysis is strongly be inhibited compared to that of a free enzyme (Fig. 1c and Supplementary Fig. 6). As a comparison, the defective structure in ZIF-8-B, well explored by XAFS and iDPC-STEM (Fig. 3 and Supplementary Figs. 13–16), can lead to the large opening pore, which favors for the catalytic substrate entrance that accelerates the biocatalysis (Fig. 1c and Supplementary Fig. 6).

**Phase transformation**. The cryo-EM, iDPC-STEM and XAFS data provided an insight into the structure-activity relationship of ZIFs biocomposites under different crystallization pathways that previously not understood. We also discovered an interesting phase transformations in BZIF-8-B, but not in BZIF-8-S. (Fig. 5a). When prolonging the crystallization time to 7 d, the original *SOD*-type biocrystal of BZIF-8-B could transform into a *dia*-type biocrystal with sheet nanoarchitecture (Fig. 5b and Supplementary Fig. 20), which is a thermodynamically stable phase[30]. Whereas this phase transformation process could not been observed in BZIF-8-S with highly crystalline structure (Fig. 2), even though the crystallization time was extending to 2 months (Supplementary Fig. 21). Such phase transformation process of BZIF-8-B may result from the crystallographic instability, because BZIF-8-B contained a mass of amorphous phase and coordination defected units (Fig. 3).

We next focused on the structure-activity relationship after phase transformations. The lattice-resolution cryo-EM images from [110] projection showed that the *dia*-type BZIF-8-B possessed a high crystalline structure, wherein the vertically

arranged lattice was recorded throughout the framework (Fig. 5c and Supplementary Fig. 22). In addition, the intensity of the small-angle X-ray scattering (SAXS) at $q$ ranged from 0.1 to 0.4 Å$^{-1}$, assigned to the microporous region[19], was significantly decreased after phase transformations (Fig. 5d). This suggested that the *dia*-type BZIF-8-B has less microporosity compared with the original BZIF-8-B. The CLSM experiment, wherein the incorporated proteins were labelled with green fluorescence, was carried out to explore the spatial distribution of proteins after phase transformation. In the original BZIF-8-B biocomposites, the green fluorescence completely overlaid with the crystal (Supplementary Fig. 23a), suggesting the uniformly spatial distribution of proteins. However, after phase transformations, the retained green fluorescence could not cover well with the transformed crystal, elucidating that the spatial distribution of proteins turned into uneven (Supplementary Fig. 23b). As a result, the uneven proteins spatial distribution, as well as the high crystallinity and less microporosity after phase transformation lead to the weak bioactivity of *dia*-type BZIF-8-B (Fig. 5e). This phase transformation also implied that prolonging the assembly time could trigger the microstructure change of ZIF-8 biocomposites, and hence decreased the activity.

**Stability and protecting effect**. The different microstructures of BZIF-8 biocomposites inspired us to explore their structural stability. ZIF-8 is acid-sensitive, and such nature has afforded it as protein capsules for the acid microenvironment-responsive delivery[14,15,31]. The acid-responsive release of target proteins was related to the microstructure of ZIF-8. We traced the time-dependent release of proteins in BZIF-8-S and BZIF-8-B in pH = 6 phosphate buffer (Fig. 6a). BZIF-8-B could realize 90% of the release of encapsulated proteins within 10 min, while only 28% proteins was released in BZIF-8-S after 60 min. In addition, when decreasing the pH of PBS to 5, BZIF-8-S only released 45% proteins after 40 min (Supplementary Fig. 24). The distinct releasing behavior between BZIF-8-S and BZIF-8-B can be well explained by the dissected microstructure above. BZIF-8-S was highly crystalline (Fig. 2), allowing the relatively high structural stability against acid. Whereas, BZIF-8-B was a multiphase structure, which existed a mass of amorphous phase and

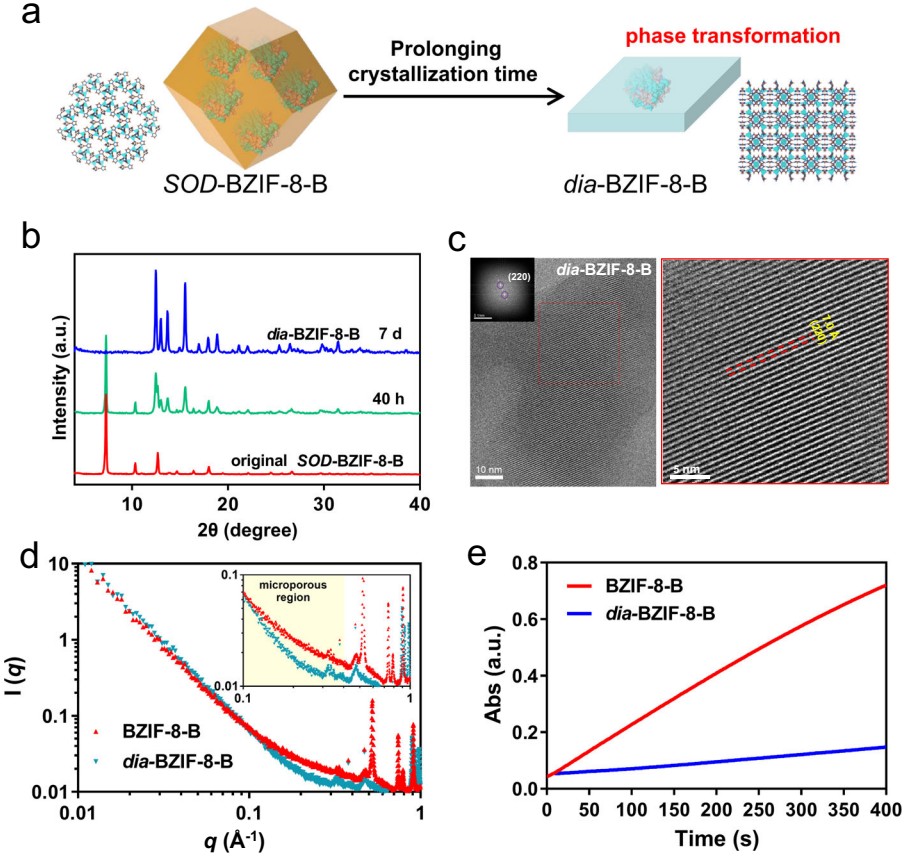

**Fig. 5 Phase transformation process. a** Schematic illustration of the phase transformation of *SOD*-type BZIF-8 when prolonging the crystallization time. **b** PXRD patterns of the fresh-prepared BZIF-8-B crystals, and the crystals when prolonging the crystallization time to 40 h and 7d. **c** The cryo-EM image showed the highly crystalline structure of *dia*-BZIF-8-B. **d** The SAXS pattern of BZIF-8-B and *dia*-BZIF-8-B. **e** The bioactivity of BZIF-8-B and *dia*-BZIF-8-B in terms of identical enzymes dosage (1 μg GOx in each material).

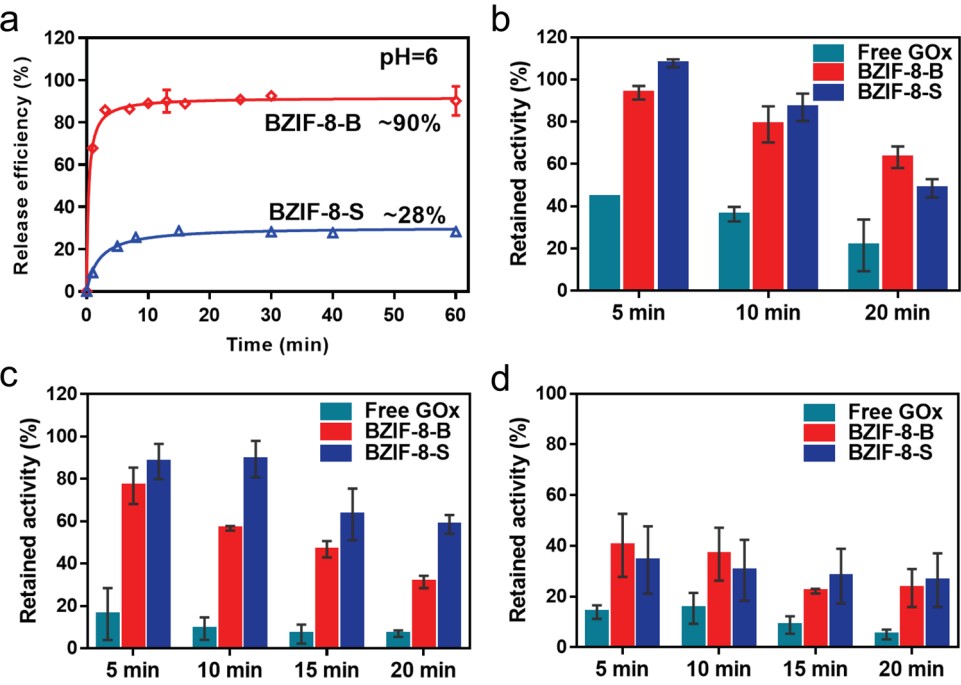

**Fig. 6 Structure-dependent stability and protecting effect. a** The time-dependent release of proteins of BZIF-8-B and BZIF-8-S at pH = 6 PBS. Error bars = Standard Deviation (*n* = 3). The retained activities of free GOx, BZIF-8-B, and BZIF-8-S after heating (**b**), 10 M urea (**c**) and acetone (**d**) treatment for different times. Error bars (SD) are presented in each group. Error bars = Standard Deviation (*n* = 3).

coordination defected units (Fig. 3). The defective MOFs structure usually shows lower chemical stability compared to the highly crystalline analogue[32], and without the compensating groups in the defective pore, the extent of degradation in the defective MOFs increases as the concentration of defects in the material increases. Therefore, the crystallographic structure of BZIF-8-B was more sensitive to acid conditions. These findings may guide to design protein capsule with different response behaviors.

We next investigated the protecting effect of the ZIF-8 shell on the guest proteins. The BZIF-8-S and BZIF-8-B were exposed to the urea, high temperature and organic solvent, respectively[33,34]. The fragile proteins should been denatured in these harsh conditions. Urea and high temperature can disturb the H-bonded network of proteins, and devitalizes the proteins through the unfolding effect. The highly crystalline network in BZIF-8-S favored for the tightly structural confinement of the guest proteins. Indeed, as shown in Figs. 6b, c. BZIF-8-S showed a better protecting effect than BZIF-8-B toward guest proteins at high-temperature (85 °C) and 10 M urea conditions. However, the protecting effects between BZIF-8-B and BZIF-8-S were similar at organic solvent, and the retained bioactivities after organic solvent treatment were lower than those after high temperature and urea treatments (Fig. 6d). This phenomenon was caused by the fact that organic solvent molecules denature proteins not only by unfolding effect but also by penetrating into the active sites and changing the local environment[35,36].

## Discussion

We unveil the refined structure of the next-generation biomaterials, BZIF-8, using the atomic-level characterization of cryo-EM, iDPC-STEM, and XAFS techniques. Based on these informations, we can discover an obvious difference on nanoarchitecture under different crystallization pathways. The BZIF-8, crystallized by solid-state transformation, is highly crystalline. The 6-ring window (along the [111] projection) and 4-ring window (along the [100] direction) of ZIF-8 are 3.4 Å and 2.8 Å (Fig. 2), respectively. Such narrow opening windows inevitably limit the catalytic substrate diffusion and product transport. Thus, the highly crystalline structure of BZIF-8-S presents decreasing activity. Whereas in the case of crystallization by biomacromolecules-triggered crystallization, the obtained BZIF-8 is a multiphase structure, which coexisted a mass of crystalline phase, amorphous phase and coordination defected units. Such different microstructure between BZIF-8-S and BZIF-8-B may result from the different crystallization pathway and rate (Supplementary Figs. 1 and 2). As shown in the high-resolution structure (Fig. 3), the amorphous phase and coordination defected units result in the larger opening pores that favors for catalytic substrate entrance and product transport. Therefore, the multiphase structure of BZIF-8-B maintains high enzymes activity, but it will compromise to the structural stability because of the thermodynamic instability of defects (Fig. 6a). Beside, an interesting phase transformation process was found when further prolonging the crystallization time. This phase transformation-induced microstructure change also prominently affects the bioactivity.

This work provides an important insight into the structure-activity relationship of ZIF-8 biohybrid synthesized in different scenarios, and may give a reasonable explanation on the confused bioactivity observed previously. We envisage that these microstructure profiles will also be useful for the investigation of other frameworks biocomposites such as biomacromolecule-covalent organic frameworks[37] or biomacromolecule-hydrogen-bonded organic frameworks[38,39] that are sensitive to electron beams.

## Methods

**Reagent and materials**. All chemicals and reagents were purchased from commercial sources and used without further purification. Glucose oxidase (GOx, from Aspergillus niger, >180 U/mg), horseradish peroxidase (HRP, from horseradish, >300 U/mg), alcohol dehydrogenase (ADH, from Saccharomyces cerevisiae, >300 U/mg), catalase (CAT, form bovine liver, ≥200 U/mg), glucose (AR), hydrogen peroxide ($H_2O_2$, AR) and urea (AR, 99%) were purchased from Aladdin Chemistry Co., Ltd. (Shanghai, China). Zinc acetate dihydrate ($ZnAc_2 \cdot 2H_2O$, 99%), 2-methyl imidazole (HmIM, 99%), and 3,3′,5,5′-Tetramethylbenzidine (TMB, 99%) were purchased from J&K Scientific (Beijing, China). Fluorescein isothiocyanate (FITC, 96%) was purchased from Macklin Inc. (Shanghai, China).

**Characterization**. The ultraviolet-visible (UV-Vis) absorbance measurement was performed with a 2800 S spectrophotometer (SOPTOP, Shanghai). Powder X-ray diffraction (PXRD) patterns were collected (0.02°/step, 0.12 seconds/step) on a Bruker D8 Advance diffractometer (Cu Kα) at room temperature. $N_2$ adsorption isotherms were collected with a JW-DX Surface Area Analyzer at −196 °C. All the samples were pretreated under 100 °C for 12 h before measurements. Fourier transform infrared (FT-IR) spectroscopy was carried out on Bruker EQUINOX 55 spectroscopy using the ATR mode (32 scans in the 4000–400 $cm^{-1}$ spectral range). Thermogravimetric analyses (TGA) were performed under $N_2$ atmosphere (20 mL·$min^{-1}$) with temperature increasing at 10 °C·$min^{-1}$ using a TA-Q50 system. The samples were immersed in EtOH for 24 h and dried in vacuo at 100 °C for 12 h before TGA analysis. The morphology images of the crystals were analyzed by a SU8010 ultra-high revolution field emission scanning electron microscope (SEM, Hitachi, Japan). The spatial distribution of dye-labelled protein was profiled by confocal laser scanning microscopy (Zeiss LSM780, Carl Zeiss Meditec AG, Jena, Germany). Cryo-EM were performed on a FEI Titan Krios G3i (D3845) TEM operated at 300 kV and equipped with an autoloading mechanism, and the images were taken at a nominal magnification of 215,000 with a pixel size of 0.56 Å by 0.56 Å. The high-resolution iDPC-STEM images were obtained under a double Cs-corrected STEM (Thermofisher Scientific, Themis Z) operated at 300 kV with a convergence semi-angle of 11 mrad. The STEM was equipped with a SCORR spherical aberration corrector for the electron probe, which was aligned using a cross grating standard sample before measurements. The X-ray absorption spectra (XAS), including X-ray absorption near-edge structure (XANES) and extended X-ray absorption fine stucture (EXAFS) of the sample at K-edge was colleted at the Beamline of TLS07A1 in National Synchrotron Radiation Research Center (NSRRC), Taiwan.

*Crystallization of BZIF-8 trough biomacromolecules-induced crystallization (BZIF-8-B)*. In the low HmIM: Zn ratio, the crystallization of BZIF-8 gone through by biomacromolecules-induced crystallization[22]. Briefly, a certain amount GOx (1 mg, 2 mg, 4 mg, 6 mg and 8 mg) were dissolved in 200 µL deionized water, followed by adding 2 mL 40 mM zinc acetate. After stirring for 1 min, 2 mL 160 mM HmIM was added. The precipitate was immediately formed, and aged at room temperature for 1 h. The formed BZIF-8-B crystal was collected by centrifugation at 9166 × g for 3 min, and washed by deionized water for 3 times, and then dried under vacuum at room temperature overnight.

*Crystallization of BZIF-8 trough solid-state transformation (BZIF-8-S)*. In the high HmIM: Zn ratio, the crystallization of BZIF-8 gone through by solid-state transformation[22]. Briefly, a certain amount GOx (1 mg, 2 mg, 4 mg, 6 mg and 8 mg) were dissolved in 200 µL deionized water, followed by adding 2 mL 0.1 M zinc acetate. After stirring for 1 min, 2 mL 1.2 M HmIM was added. The mixed system kept transparent after adding HmIM, and was stirred at room temperature for 1 h that allowing the crystal formation. The formed BZIF-8-S crystal was collected by centrifugation at 9166 × g for 3 min, and washed by deionized water for 3 times, and then dried under vacuum at room temperature overnight.

*cryo-EM*. The cryo-EM imaging was performed according to the method reported by our group[38]. In briefly, the as-synthesized BZIF-8 crystals were dispersive in ethanol, and then mounted on a carbon-coated TEM grid. The specimen was dropped in liquid nitrogen, and then loaded into the microscope by a cryo-transfer loader. The imaging experiments were carried out on a Titan Krios G3i electron microscope (Thermofisher Scientific), and the operating voltage was set at 300 kV. The images were collected by a K3 Summit direct electron detector, which was equipped with a GIF Quantum energy filter (slit width 20 eV) in the counting mode (Bin 0.5). Data acquisition was performed using SerialEM 3.8. The images were taken at a nominal magnification of 215,000 with a pixel size of 0.56 Å by 0.56 Å. The optimized total dose rate was approximately 30 e−/Å2 for each micrograph. Motion-Corr2 with 2 × 2 binning was used for the motion correction of the acquired images, and all image processing steps were operated by nondose-weighted sum of all frames from each movie. The lattice spacing analysis was conducted by DigitalMicrograph (Gatan) software.

*iDPC-STEM*. The iDPC-STEM experiment was operated on a double Cs-corrected Themis Z scanning transmission electron microscope (Thermofisher Scientific), which is equipped with a SCORR spherical aberration corrector for the electron

probe. Before imaging, the electron probe was aligned using a cross grating standard sample. The imaging process was carried out at 300 kV, and the convergence semi-angle was 11 mrad. The aberration coefficients were set as following: $C_1 = 2.52$ nm, $A_1 = 1.70$ nm, $A_2 = 75.6$ nm, $B_2 = 74.4$ nm, $C_3 = 820$ nm, $A_3 = 278$ nm, $S_3 = 168$ nm, $A_4 = 4.71$ μm, $D_4 = 3.16$ μm, $B_4 = 6.02$ μm, $C_5 = 558$ μm, and $A_5 = 83.4$ μm. In order to reduce the low-frequency information in the image, four images, acquired by a segmented DF4 detector with high-path filter, were integrated. The electron-beam current used herein was lower than 1 pA, and the electron dose during the iDPC-STEM imaging was ca. 60 $e^-/Å^2$. The collection angle for DF detector and HAADF detector were set as 5~19 mrad and 20~120 mrad, separately. The lattice spacing analysis was conducted by DigitalMicrograph (Gatan) software.

*XAFS*. The X-ray absorption spectra (XAS) including X-ray absorption near-edge structure (XANES) and extended X-ray absorption fine structure (EXAFS) of the sample at K-edge were collected at the Beamline of TLS07A1 in National Synchrotron Radiation Research Center (NSRRC), Taiwan, where 1.5 GeV a pair of channel-cut Si (111) crystals was used in the monochromator. The fitting analysis was processed using software ATHENA and ARTEMIS[40]. Wavelet transform was calculated and plotted based on morlet function[41].

**Preparation of BZIF-8-S with large particle size**. For evaluating the effect of large particle size on the activity of BZIF-8, we also prepared the BZIF-8-S with a large particle size (ca. 2 μm). The synthesis procedure was similar to that of the ca. 200 nm BZIF-8-S, except that the crystallization was prolonged from 1 h to 6 h.

**Measurement of the enzyme content**. The loading content of enzymes within BZIF-8 was measured by examining the concentration differences of enzymes in the supernatant before and after encapsulation via Bradford proteins assay[42]. Typically, 20 μL enzymes sample was added into a 96-well plates. Then, 200 μL Coomassie Brilliant Blue G-250 reagent was added, and the mixed solution was incubated for 5 min at room temperature. Finally, the solution was collected and detected by UV-vis spectrophotometer. The concentration of the enzyme is proportional to the absorption strength at 595 nm.

**Examinations of enzymatic activities**

*Examination of enzymatic activities of GOx*. The enzyme activities of GOx was evaluated through tracing the production of $H_2O_2$ (Eq. 1) based on a TMB-$H_2O_2$-HRP enzymatic assay[43].

$$\text{Glucose} + O_2 \xrightarrow{\text{GOx}} \text{Gluconicacid} + H_2O_2 \tag{1}$$

In each trial including free GOx and GOx-loaded BZIF-8, the GOx dosage was kept at 6 μg, unless otherwise statement. The free GOx or BZIF-8 was dispersed into 100 μL phosphate buffer solution (pH = 7.4), followed by introducing 200 μL TMB solution containing 10 μg HRP. Then 200 μL glucose (10 mM) was added to activate the reaction. Finally, the produced $H_2O_2$ oxidized TMB to oxTMB, which generated a blue color signal. This signal, reflecting the $H_2O_2$ concentration, could be traced at 650 nm by UV-vis spectrophotometer using a time-scanning mode.

*Examination of enzymatic activities of ADH*. ADH reduces acetaldehyde to ethanol in the presence of coenzyme nicotinamide adenine dinucleotide (NADH)[44] (Eq. 2):

$$\text{Acetaldehyde} + \text{NADH} \xrightarrow{\text{ADH}} \text{Ethanol} + \text{NADH}^+ \tag{2}$$

The enzymatic activities of ADH were measured through the concentration change of NADH using a Micro Alcohol Dehydrogenase (ADH) Assay Kit (Solarbio Life Sciences, Beijing), wherein the dosage of ADH in each trail was kept the same (20 μg). The concentration change of NADH was detected at 340 nm by UV-vis spectrophotometer.

*Examination of enzymatic activities of CAT*. CAT catalyzes the decomposition of hydrogen peroxide into water and oxygen[45] (Eq. 3):

$$2H_2O_2 \xrightarrow{\text{CAT}} O_2 + 2H_2O \tag{3}$$

The enzymatic activity of CAT was measured by the time-dependent production of oxygen, and the dosage of CAT in each trail was kept the same (10 μg). Briefly, the BZIF-8 particle was dispersed into 5 mL PBS (pH = 7.4), and then 50 μL 200 mM $H_2O_2$ was added to initiate the reaction. The time-dependent production of oxygen was recorded using a dissolved oxygen analyzer.

**The effect of the proteins on BZIF-8 growth rates**. The BZIF-8 growth rates was evaluated by the transmittance change of the solution after mixing proteins and the ZIF-8 the precursors. The transmittance of the mixed solution was monitored by UV-vis absorption spectroscopy at 595 nm.

**Fluorescence labeling**. The enzymes fluorescence labeling were carried out based on the chemical conjugation between the amino of lysine residue of enzymes and the thiocarbmide of fluorescein isothiocyanate (FITC, a green fluorescence dye). In

brief, 20 mg GOx was dispersed into 10 mL carbonate buffer solution (pH = 9.0, 0.5 M), followed by adding 1 mg FITC. The mixed solution was then stirred for 5 h in dark conditions. Finally, the FITC-labelled GOx were obtained through ultrafiltration by a centrifugal filter device (molecular weight cut-off MWCO = 8 kDa) for 3 times to remove excess reaction reagents and salts

**Phase transformation experiment**. For the phase transformation experiment, the crystallization time was prolonged to 40 h and 7d, and the precipitate was collected for structural analysis. In order to profile the spatial distribution of protiens during this phase transformation, the proteins were pre-labelled with FITC. The as-synthesized BZIF-8-B crystal (1 h, *SOD*-type) and the phase-transformed *dia*-BZIF-8-B (7d) was collected for CLSM analysis.

**Proteins release experiment**. For the proteins release experiment, FITC-proteins were used in the crystallization of BZIF-8-S and BZIF-8-B. The obtained BZIF-8-S and BZIF-8-B were dispersed in pH = 6 or PH = 5 PBS solution, respectively. After the different time of incubation, the supernatant was collected, and the release of proteins was quantified by fluorescence intensity at 520 nm (488 nm excitation wavelength).

## Data availability

All data supporting this study and its findings are available within the article and its Supplementary Information or from the corresponding authors upon request.

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

## Acknowledgements

We thank for the Shanghai Nanoport of Thermo Fisher Scientific (Shanghai, China) for the iDPC-STEM imaging. We acknowledge financial support from projects of the National Natural Science Foundation of China (22174164, G.C.; 22104159, S.H.; 21904146, G.C.), Natural Science Foundation of Guangdong Province (2020A1515010825, G.C.; 2019A1515011722, S.H.) and the Fundamental Research Funds for the Central Universities (2021qntd24, G.C.).

## Author contributions

G.C. conceived the idea, designed the experiments, wrote the manuscript and provided financial support. L.T. and S.H. performed material synthesis and characterization. Y.S. helped with the material synthesis, characterization and data analysis. S. L. assisted with the iDPC-STEM imaging. X.M. performed the Cryo-EM experiment and helped with the data analysis. F.Z. participated in the discussions. G.O. supervised the experiments and provided financial support.

## Competing interests

The authors declare no competing interests.
