## [Peer Review File · Nature Communications]

Reviewers' comments:

Reviewer #1 (Remarks to the Author):

Since the report of encapsulating enzyme in ZIF-8 (One-pot synthesis of protein-embedded metal-organic frameworks with enhanced biological activities. *Nano Letters* 2014, 14, 5761-5765), this method has been widely used by many others to stabilize enzyme for various applications. Although the stability of enzyme in ZIF-8 was observed to be largely enhanced by different studies, the apparent activities of enzyme in ZIF-8 were different from case to case. It is very necessary to understand the structure-activity relationship. The authors use cryo-electron microscopy to investigate the structure of enzyme-ZIF-8 composites which were synthesized at two different conditions, high concentration of precursors and low concentration of precursors. The authors found that in the case of low concentration of precursors, the formation of ZIF-8 was induced by protein and some defects and amorphous structure were formed and therefore the apparent activity of enzyme is high due to the less substrate transportation limitation. This finding is extremely important for future development of this method and can provide important guidance for others to design the enzyme-ZIF-8 composites. The manuscript can be accepted for publication after addressing the following points:

Ge and coworkers pioneered in using the cryo-electron microscopy to investigate the amorphous structure and mesopores in enzyme-ZIF composites and proposed the defects in structure are very important for retaining high enzyme activity (ref. Packaging and delivering enzymes by amorphous metal-organic frameworks. *Nature Communications* 2019, 10, 5165; Defect-induced activity enhancement of enzyme-encapsulated metal-organic frameworks revealed in microfluidic gradient mixing synthesis. *Science Advances* 2020, 6, eaax5785). In this study, the authors did a very excellent job of using cryo-electron microscopy to reveal the atomic level structure to systematically demonstrate the role of defects. I suggest the authors to mention the above literature to discuss the relationship between current study and previous studies.

What is difference in enzyme loading percentage for two different methods, BZIF-8-S and BZIF-8-B?

The relatively low stability of enzyme in BZIF-8-B was due to the defects? Please discuss more on this point.

Reviewer #2 (Remarks to the Author):

The authors describe a TEM and synchrotron x-ray study of the structure property relationships for biomacromolecule metal-organic framework composites. Specifically, the authors synthesize Glucose oxidase ZIF-8 under different conditions and evaluate the enzymatic activity using the TMB-H₂O₂-HRP enzymatic assay. The authors then characterize the crystals using HR-TEM, iDPC-STEM and XAS.

Overall, I find this study highly novel and impactful. The experiments have been performed to a high level and provide novel insights into the structure property relationships for these materials, which is a much-needed piece of information. However, there are some key pieces of information missing before I can recommend publication in Nature Communications.

- 1) As a general comment the main pieces of information missing are the “why’s” at present the authors present very nice data but do not discuss why the crystals are structurally different (based on the mechanism) or why the crystals have different activity (based on the mechanism of catalysis). This leaves the paper lacking of the key piece of information needed to make this high impact. I would suggest the authors include a discussion section which provides a “why” explanation to the data and provides a guide for researchers to know how to optimize their synthesis conditions.
- 2) It is unclear exactly which samples are being compared. The authors use a range of different starting protein concentrations and obtain a range of different encapsulation efficiencies. The authors should state exactly which samples are used for the characterization studies
- 3) As the encapsulation efficiencies are not the same. How was the activity assay made “fair”? Are the authors comparing samples with the same concentration of enzyme?
- 4) The authors claim that BZIF-8-B is more crystalline and that BZIF-8-S contains an amorphous phase. It is very interesting to note that this amorphous phase does NOT show up in the XRD. XRD is commonly used to determine crystallinity in MOFs and this is a good example of why bulk averaging data is insufficient. It is also important for the authors to speculate on why the XRD cannot pick this up.
- 5) The authors should speculate on why BZIF-8-B is more crystalline and BZIF-8-S contains an amorphous phase considering both mechanisms go through an amorphous precursor phase
- 6) The authors note that the Zn coordination numbers are different between why BZIF-8-B and BZIF-8-S, but there is no description of why this is the case or why this influences the enzymatic activity. This should be included.
- 7) In the SI the authors show SEM images of the two types of crystals. These show very different nano and microscale structures. Does this also have an effect on the activity? In addition to the higher resolution information discussed in the paper?

8) Minor comment but the general format for describing biomacromolecules-metal-organic frameworks is B@MOF or Protein-metal-organic frameworks (P@MOF). This is to distinguish them from structures where the biomacromolecule is a part of the framework.

Reviewer #3 (Remarks to the Author):

This manuscript deals with the studies of local structure of MOF-protein composites. Authors prepared two series of MOF-protein samples by using different concentration regimes. These two series differ in overall crystallinity, protein content, and enzyme activity. Authors used cryo-electron microscopy to examine ordering in the MOF samples and found significant number of defects for MOFs prepared with low concentration of Zn and a ligand. Unfortunately, the experimental part of the paper does not contain enough details (both the text and ESI). For example, specific workup, purification, and washing procedures for obtained MOF samples are not given. Yields of MOFs are not given. Authors provide content of a protein in MOF in the Table S-1, demonstrating much higher loads in the case of the “low concentration” preparation route. However, as no information is given on the yields of MOFs, the reader cannot comprehend the absolute amount of the protein that was encapsulated in MOFs. Experimental part should be fixed and described in full detail.

This paper appears to be complimentary to a recent publication in JACS (J. Am. Chem. Soc. 2020, 142, 1433–1442) that discussed the same approach (low and high concentration regimes) for the synthesis of the same MOF (ZIF-8) in the presence of a protein (BSA, different from this submission). In that JACS paper authors also used cryoTEM, among other techniques, in order to study the process of nucleation and growth of protein-MOF composites. One finding of that paper is that the low-concentration regime produces MOF crystals with large nanometer-sized pores/defects. It appears that this submission is in line with those previous literature findings. Overall, this is complimentary work on a similar system, which is in agreement with a general consensus in this community. As such, this manuscript is better suited for more specialized audience and possibly JACS is an appropriate target journal.

Response to the reviewers' comments

Reviewer #1 (Remarks to the Author):

Since the report of encapsulating enzyme in ZIF-8 (One-pot synthesis of protein-embedded metal-organic frameworks with enhanced biological activities. Nano Letters 2014, 14, 5761-5765), this method has been widely used by many others to stabilize enzyme for various applications. Although the stability of enzyme in ZIF-8 was observed to be largely enhanced by different studies, the apparent activities of enzyme in ZIF-8 were different from case to case. It is very necessary to understand the structure-activity relationship. The authors use cryo-electron microscopy to investigate the structure of enzyme-ZIF-8 composites which were synthesized at two different conditions, high concentration of precursors and low concentration of precursors. The authors found that in the case of low concentration of precursors, the formation of ZIF-8 was induced by protein and some defects and amorphous structure were formed and therefore the apparent activity of enzyme is high due to the less substrate transportation limitation. This finding is extremely important for future development of this method and can provide important guidance for others to design the enzyme-ZIF-8 composites. The manuscript can be accepted for publication after addressing the following points:

Response: We thank for the reviewer's positive comments.

Ge and coworkers pioneered in using the cryo-electron microscopy to investigate the amorphous structure and mesopores in enzyme-ZIF composites and proposed the defects in structure are very important for retaining high enzyme activity (ref. Packaging and delivering enzymes by amorphous metal-organic frameworks. Nature Communications 2019, 10, 5165; Defect-induced activity enhancement of enzyme-encapsulated metal-organic frameworks revealed in microfluidic gradient mixing synthesis. Science Advances 2020, 6, eaax5785). In this study, the authors did a very excellent job of using cryo-electron microscopy to reveal the atomic level structure to systematically demonstrate the role of defects. I suggest the authors to mention the above literature to discuss the relationship between current study and previous studies.

Response: We thank for the reviewer's kindly suggestions. We appreciate the works reported by Ge and coworkers, also cited them in our manuscript. They utilized defective engineering to enhance the enzyme activity of enzyme@ZIF-8 biocomposites. In previous studies, these defects are usually characterized by indirect methods. Even though using the advanced cryo-electron microscopy, it is still unable to directly image the high-resolution structure of enzyme@ZIF-8, because of the limited resolution and high signal-to-noise ratio.

In the current study, we focused on the high-resolution structure of enzyme@ZIF-8 and the deep understanding of structure-activity relationship. The atomic-level structure information of enzyme@ZIF-8, crystallized in different pathways, are directly imaged and identified for the first time. We captured the defective structure by iDPC-STEM technique (rather cryo-EM), and demonstrated the role of assembling conditions on crystalline, defective and amorphous structure of an enzyme@ZIF-8.

Ge's excellent works demonstrate that the defects in structure are very important for retaining high enzyme activity, which may provide a deeper insight for the key role of defective structure observed in our study. We have given more discussion in the revised manuscript (page 8, line 178-180).

What is difference in enzyme loading percentage for two different methods, BZIF-8-S and BZIF-8-B?

Response: The enzyme loading percentage for two different methods is displayed in Table S1. In general, BZIF-8-B showed higher enzyme loading than BZIF-8-S, because of the different nucleation

modes.

The relatively low stability of enzyme in BZIF-8-B was due to the defects? Please discuss more on this point.

Response: Previous research has showed that the defective MOFs structure usually has lower chemical stability compared to the highly crystalline analogue, and without the compensating groups in the defective pore, the extent of degradation in the defective MOFs increases as the concentration of defects in the material increases (J. Phys. Chem. C 2017, 121, 23471–23479). We believe that the relatively low stability of enzyme in BZIF-8-B was due to the defects, which was well-explored by the iDPC-STEM and XAFS. We have given more discussion in the revised manuscript (page 13, line 275-279).

Reviewer #2 (Remarks to the Author):

The authors describe a TEM and synchrotron x-ray study of the structure property relationships for biomacromolecule metal-organic framework composites. Specifically, the authors synthesize Glucose oxidase ZIF-8 under different conditions and evaluate the enzymatic activity using the TMB-H₂O₂-HRP enzymatic assay. The authors then characterize the crystals using HR-TEM, iDPC-STEM and XASf.

Overall, I find this study highly novel and impactful. The experiments have been performed to a high level and provide novel insights into the structure property relationships for these materials, which is a much-needed piece of information. However, there are some key pieces of information missing before I can recommend publication in Nature Communications.

Response: We thank for the reviewer's positive comments.

1) As a general comment the main pieces of information missing are the “why’s” at present the authors present very nice data but do not discuss why the crystals are structurally different (based on the mechanism) or why the crystals have different activity (based on the mechanism of catalysis). This leave the paper lacking of the key piece of information needed to make this high impact. I would suggest the authors include a discussion section which provides a “why” explanation to the data and provides a guide for researchers to know how to optimize their synthesis conditions.

Response: We thank for the reviewer's meaningful suggestion. In the Discussion section, we have provided the detail discussion on the structure and activity difference of the enzyme@ZIF-8 crystal.

The BZIF-8, crystallized by solid-state transformation, is highly crystalline because. The 6-ring window (along the [111] projection) and 4-ring window (along the [100] direction) of ZIF-8 are only 3.4 Å and 2.8 Å (Figure 2), respectively. Such narrow opening windows inevitably limit the catalytic substrate (glucose, ca. 4.0 Å) diffusion and product transport. Thus, the highly crystalline structure of BZIF-8-S presents decreasing activity. Whereas, in the case of crystallization by biomacromolecules-triggered crystallization, the obtained BZIF-8 is a multiphase structure, which coexisted a mass of crystalline phase, amorphous phase and coordination defected units. Such different microstructure between BZIF-8-S and BZIF-8-B may result from the different crystallization pathway and rate (Figure S1 and S2). As shown in the high-resolution structure (Figure 3), the amorphous phase and coordination defected units result in the larger opening pores that favors for catalytic substrate entrance and product transport. Therefore, the

multiphase structure of BZIF-8-B maintains high enzymes activity, but it will compromise to the structural stability because of the thermodynamic instability of defects (Figure 6a).

We have provided these discussion in the Discussion section (highlighted in yellow), and can provide a clear guide for optimizing the synthesis conditions.

2) It is unclear exactly which samples are being compared. The authors use a range of different starting protein concentrations and obtain a range of different encapsulations efficiencies. The authors should state exactly which samples are used for the characterization studies.

Response: We are sorry for this mistake. All of the BZIF-8 samples used for the cryo-EM, iDPC-STEM and XAFS characterization studies are crystallized using 8 mg enzymes, unless otherwise statement. We have stated it in the revised manuscript.

3) As the encapsulation efficiencies are not the same. How was the activity assay made “fair”? Are the authors comparing samples with the same concentration of enzyme?

Response: The encapsulation efficiencies are different since the different crystallization pathways. In order to make the activity assay “fair”, the concentration of enzyme in each activity assay of free enzymes, BZIF-8-B and in BZIF-8-B was kept the same. We have highlighted this in the Manuscript Text and the Supplementary Methods section.

4) The authors claim that BZIF-8-B is more crystalline and that BZIF-8-S contains an amorphous phase. It is very interesting to note that this amorphous phase does NOT show up in the XRD. XRD is commonly used to determine crystallinity in MOFs and this is a good example of why bulk averaging data is insufficient. It is also impost for the authors to speculate on why the XRD cannot pick this up.

Response: XRD is a widely used method to explore the crystallinity of the materials. However, the XRD reflects the macroscopic phase of the material, and is unable to clearly identify the specifically detective and amorphous phases in the crystallization system. Seeing is believing. TEM techniques, including cryo-EM and iDPC-STEM, can directly image the microstructure with high-resolution, which can't be obtained by PXRD. We have given more discussion in the revised manuscript (page 6, line 115-121)

5) The authors should speculate on why BZIF-8-B is more crystalline and BZIF-8-S contains an amorphous phase considering both mechanisms go through an amorphous precursor phase

Response: We think the different crystallinity of BZIF-8-B and BZIF-8-S may result from the different crystallization rate. For BZIF-8-B, the biomacromolecules trigger the ZIF-8 crystallization around their surfaces, and the crystallization rate is fast (Figure S1) that results in the relatively poor crystallinity. However, for BZIF-8-S, the biomacromolecules amorphous phase undergoes a solid-state transformation at the spontaneously growing ZIF-8 crystal surface through heterogeneous crystallization. Such crystallization rate is relatively slow (Figure S2), and hence favors for the highly crystalline species growth. We have provided the explanation in the revised manuscript (page 8, line 171-177).

6) The authors note that the Zn coordination numbers are different between why BZIF-8-B and BZIF-8-S, but there is no description of why this is the case or why this influences the enzymatic activity. This should be included.

Response: Base on the fitted R-space data, we found that the apparent local coordination number of the Zn atoms in BZIF-8-S is 4.06 (Figure 4b), which was well in agreement with the theoretical coordination

number in highly crystalline SOD-type ZIF-8. However, the calculated coordination number of the Zn atoms in BZIF-8-B turned out to be 4.272 (Figure 4b). The higher apparent coordination number of Zn atoms indicated that the loss of Zn atoms, and also suggested the defective structure that well identified by the iDPC-STEM (Figure 3).

It is observed that the 6-ring (along the [111] projection) and 4-ring (along the [100] direction) opening windows of ZIF-8-S are only 3.4 Å and 2.8 Å (Figure 2), respectively, which will limit the diffusion of catalytic substrates. The defective structure in ZIF-8-B, well explored by XAFS (Figure 4) and iDPC-STEM (Figure 3), can lead to the large opening pore, which favors for the catalytic substrate entrance that accelerates the biocatalysis.

We have provide more discussion in the revised manuscript (page 10, line 214-218).

7) In the SI the authors show SEM images of the two types of crystals. These show very different nano and microscale structures. Does this also have an effect on the activity? In addition to the higher resolution information discussed in the paper?

Response: We thank for the reviewer's meaningful comments. To exclude the effect of particle size on activity, we prepared the ZIF-8-S with different size by adjusting the crystallization time (Figure S9a). The results showed that the particle size has limited effect on the activity (Figure S9b).

Figure S9. The SEM images (a) and the bioactivity (b) of prepared BZIF-8-S with ca. 200 nm and 2 μm. The enzymes (GOx) dosage for the activity test in different material were kept the same (14 μg).

8) Minor comment but the general format for describing biomacromolecules-metal-organic frameworks is B@MOF or Protein-metal-organic frameworks (P@MOF). This is to distinguish them from structures where the biomacromolecule is a part of the framework.

Response: Thanks for the reviewer's suggestion. We denoted biomacromolecules-zeolitic imidazolate frameworks-8 as BZIF-8 herein, because the biomacromolecules participated in crystallization of ZIF-8, particularly in the BZIF-8-B.

Reviewer #3 (Remarks to the Author):

This manuscript deals with the studies of local structure of MOF-protein composites. Authors prepared two series of MOF-protein samples by using different concentration regimes. These two series differ in overall crystallinity, protein content, and enzyme activity. Authors used cryo-electron microscopy to

examine ordering in the MOF samples and found significant number of defects for MOFs prepared with low concentration of Zn and a ligand. Unfortunately, the experimental part of the paper does not contain enough details (both the text and ESI). For example, specific workup, purification, and washing procedures for obtained MOF samples are not given. Yields of MOFs are not given. Authors provide content of a protein in MOF in the Table S-1, demonstrating much higher loads in the case of the “low concentration” preparation route. However, as no information is given on the yields of MOFs, the reader cannot comprehend the absolute amount of the protein that was encapsulated in MOFs. Experimental part should be fixed and described in full detail.

Response: Thank you for the reviewer’s meaningful comment. We are sorry for the unclear expression of the experimental details. According to the reviewer’s comment, we have provided more detail information of the experimental part in the revised manuscript. Specially, the specific workup, purification, and washing procedures for the obtained MOF samples are provided in detail in the Methods section (as described below)

Crystallization of BZIF-8 through biomacromolecules-induced crystallization (BZIF-8-B). In the low HmIM: Zn ratio, the crystallization of BZIF-8 gone through by biomacromolecules-induced crystallization²². Briefly, a certain amount GOx (1 mg, 2 mg, 4 mg, 6 mg and 8 mg) were dissolved in 200 μ L deionized water, followed by adding 2 mL 40 mM zinc acetate. After stirring for 1 min, 2 mL 160 mM HmIM was added. The precipitate was immediately formed, and aged at room temperature for 1 h. The formed BZIF-8-B crystal was collected by centrifugation at 10000 rpm for 3 min, and washed by deionized water for 3 times, and then dried under vacuum at room temperature overnight.

Crystallization of BZIF-8 through solid-state transformation (BZIF-8-S). In the high HmIM: Zn ratio, the crystallization of BZIF-8 gone through by solid-state transformation²². Briefly, a certain amount GOx (1 mg, 2 mg, 4 mg, 6 mg and 8 mg) were dissolved in 200 μ L deionized water, followed by adding 2 mL 0.1 M zinc acetate. After stirring for 1 min, 2 mL 1.2 M HmIM was added. The mixed system kept transparent after adding HmIM, and let it stand at room temperature for 1 h that allowing the crystal formation. The formed BZIF-8-S crystal was collected by centrifugation at 10000 rpm for 3 min, and washed by deionized water for 3 times, and then dried under vacuum at room temperature overnight.

In addition, the yields of enzyme@ZIF-8 and the absolute amount of the encapsulated enzyme have been provide in Table S1.

This paper appears to be complimentary to a recent publication in JACS (J. Am. Chem. Soc. 2020, 142, 1433–1442) that discussed the same approach (low and high concentration regimes) for the synthesis of the same MOF (ZIF-8) in the presence of a protein (BSA, different from this submission). In that JACS paper authors also used cryoTEM, among other techniques, in order to study the process of nucleation and growth of protein-MOF composites. One finding of that paper is that the low-concentration regime produces MOF crystals with large nanometer-sized pores/defects. It appears that this submission is in line with those previous literature findings. Overall, this is complimentary work on a similar system, which is in agreement with a general consensus in this community. As such, this manuscript is better suited for more specialized audience and possibly JACS is an appropriate target journal.

Response: We thank for the reviewer's comment. As the reviewer mentioned, this JACS work (J. Am. Chem. Soc. 2020, 142, 1433–1442) focused on the different crystallization mode of BSA@ZIF-8 through tracing the nucleation process. We also highly acknowledged this work (Ref 22 in our manuscript), because the different enzymes@ZIF-8 in our work was prepared based on the crystallization mode reported by the JACS work. We understand the potential concerns of the reviewer. However, we believe our work presents several advances that previously not been profiled, which is also affirmed by Reviewer 1 and Reviewer 2.

Firstly, the JACS work focused on the nucleation process of BSA@ZIF-8 and no structure-activity relationship was involved. The goal of our work is unveiling the high-resolution structure of enzymes@ZIF-8 and understanding the structure-activity relationship. Therefore, the topic of our work is different from the JACS work.

Secondly, as we known, the activity of enzyme@ZIF-8 is controversial, and one of the technological difficulty to resolve these controversial views is profiling the high-resolution structure of the enzyme@ZIF-8 in different preparing conditions. Such technological difficulty resulted from the electron beams-sensitive structure of ZIF-8. As far as we know, the atomic-resolution structures of enzymes@ZIF-8 are still not identified, which limits the understanding of the structure-activity relationship. As shown in the JACS work, the cryo-EM only provided the lattice-resolution structure of enzymes@ZIF-8. It only enabled to demonstrate the crystallinity of enzyme@ZIF-8, which can be well-explored by the traditional PXRD test. That is, the microstructural information could not been captured (seen in Figure 2 below). As a comparison, the atomic-resolution structure of enzymes@ZIF-8 is identified for the first time using the emerging integrated differential phase contrast-scanning transmission electron microscope (iDPC-STEM) technique combined with XAFS in our work (seen in Figure 2 below). The iDPC-STEM is a new imaging technique that enables to image the beam-sensitive materials with high-resolution and signal-to-noise ratio (Nature 2021, 592, 541–544). Based on the high-resolution structure obtained combined with iDPC-STEM and XAFS, our work can well explain the activity difference observed in enzymes@ZIF-8, which is the main innovation of our work.

lattice-resolution images (cryo-EM)

J. Am. Chem. Soc. 2020, 142, 1433–1442

Atomic-resolution images (iDPC-STEM)

This work

Figure 2. The structural resolution in JACS paper and our work

Thirdly, indeed, this JACS work found that the low-concentration regime produced MOF crystals with large nanometer-sized pores (seen in Figure 3 below). As the JACS author mentioned: “We also hypothesize that the pores in the final BSA-ZIF-8 crystals are related to the aggregation and crystallization of the irregularly shaped amorphous phase”, the pore formation in the BSA@ZIF-8 is hypothetical and unclear. Importantly, the pore environment including the opening window and phase species, which significantly affects the bioactivity of enzymes, is unable to be profiled. In our work, the advanced iDPC-STEM combined with the XAFS provided a new insight of the microstructure. We clearly imaged the high-resolution structure of enzyme@ZIF-8, and the multiphase structure, which coexisted a mass of crystalline phase, amorphous phase and coordination defected units were directly identified (seen in Figure 3 below). These structural information have not been seen previously, and could significantly influence the activity of enzymes@ZIF-8.

Pore observed in
BSA@ZIF-8 (cryo-EM)
J. Am. Chem. Soc. 2020,
142, 1433–1442

Directly identified the multiphase structure
This work

Figure 3. The pore observed in JACS paper and the multiphase structure profiled in our work.

In addition, the phase transformation process and subsequent bioactivity change were discovered in this work (Figure 5 in the manuscript). The stability and protective effect of enzymes@ZIF-8 with different microstructure were also studied (Figure 6 in the manuscript). Such new findings have not been explored before.

We understand the potential concerns of the reviewer, but we believe that the new findings are still important for future development of enzymes@MOFs biocomposites and can provide important guidance for others to design this new biomaterial. This enzymes@MOFs technique has been received increasing attentions, and shows huge potential in multidisciplinary applications including biosensing, catalysis, bio-storage and nanomedicine etc (*Nature Catalysis* 2018, 1, 689–695; *Nat. Commun.* 2019, 10, 5002; *JACS* 2018, 140, 9912–9920; *Nat. Commun.* 2015, 6, 7240). Therefore, we think our findings should be suitable for the audiences of *Nature Communications*. For clearly presenting the advance and novelty of our findings, we have given more discussion on the difference between the JACS work and our work in the revised manuscript.

REVIEWER COMMENTS

Reviewer #1 (Remarks to the Author):

The authors have addressed my concerns.

Reviewer #2 (Remarks to the Author):

The authors have successfully addressed all of my previous comments and I recommend publication as is. I strongly disagree with reviewer #3 that this provides similar information to J. Am. Chem. Soc. 2020, 142, 1433-1442. As the authors point out, focusing on differences in activity and the atomic resolution structure makes this a significant increase in our understanding of these materials.

Response to the reviewers' comments

Reviewer #1 (Remarks to the Author):

The authors have addressed my concerns.

Response: We sincerely thank reviewer #1 for his/her positive evaluation and recommendation.

Reviewer #2 (Remarks to the Author):

The authors have successfully addressed all of my previous comments and I recommend publication as is. I strongly disagree with reviewer #3 that this provides similar information to J. Am. Chem. Soc. 2020, 142, 1433-1442. As the authors point out, focusing on differences in activity and the atomic resolution structure makes this a significant increase in our understanding of these materials.

Response: We sincerely thank the reviewer #2 for his/her approbation of the innovation of our work. Thanks again for the recommendation.